# Canary Bornavirus (*Orthobornavirus serini*) Infections Are Associated with Clinical Symptoms in Common Canaries (*Serinus canaria* dom.)

**DOI:** 10.3390/v14102187

**Published:** 2022-10-04

**Authors:** Monika Rinder, Noreen Baas, Elisabeth Hagen, Katrin Drasch, Rüdiger Korbel

**Affiliations:** 1Clinic for Birds, Small Mammals, Reptiles and Ornamental Fish, Centre for Clinical Veterinary Medicine, University of Munich, 85764 Oberschleissheim, Germany; 2Institute of Sociology, Friedrich-Alexander-Universität Erlangen-Nürnberg, 91054 Erlangen, Germany

**Keywords:** avian bornavirus, canary bornavirus, domestic canary, natural infection, pathogenicity, circovirus, polyomavirus, *Macrorhabdus*, proventricular dilatation disease, coinfection

## Abstract

While parrot bornaviruses are accepted as the cause of proventricular dilatation disease (PDD) in psittacine birds, the pathogenic role of bornaviruses in common canaries is still unclear. To answer the question of whether canary bornaviruses (species *Orthobornavirus serini*) are associated with a PDD-like disease in common canaries (*Serinus canaria* f. dom.), the clinical data of 201 canary bird patients tested for bornaviruses using RT-PCR assays, were analyzed for the presence of PDD-like gastrointestinal or central nervous system signs and for other viruses (mainly circovirus and polyomavirus), yeasts and trichomonads. Canary bornavirus RNA was detected in the clinical samples of 40 out of 201 canaries (19.9%) coming from 28 of 140 flocks (20%). All nucleotide sequences obtained could unequivocally be determined as canary bornavirus 1, 2, or 3 supporting the current taxonomy of the species *Orthobornavirus serini*. PDD-like signs were found associated with canary bornavirus detection, and to a lesser extent, with circoviruses detection, but not with the detection of polyomaviruses, yeasts or trichomonads. The data indicate that canary bornaviruses contribute to a PDD-like disease in naturally infected canaries, and suggest a promoting effect of circoviruses for the development of PDD-like signs.

## 1. Introduction

Avian bornaviruses were first b described in 2008 [1,2]. They are the causative agent of a disease of parrots and other psittacines termed proventricular dilatation disease (PDD), which is characterized by gastrointestinal symptoms (diarrhea, regurgitation, digestive disorders with dilatation of the proventriculus, reduced body weight, excretion of undigested seeds in the feces) as well as central nervous signs (ataxia, seizures, tremors) and ocular disorders (disseminated chorioretinitis, impaired vision) [3,4]. The etiology has been unequivocally proven by experimental infections leading to such clinical signs in a variety of psittacines such as cockatiels, conures, or African Grey parrots [5,6,7,8]. Avian bornaviruses can, however, be found in a broad range of avian species such as waterfowl [9,10], estrildid finches [11], and common or domestic canaries, *Serinus canaria* form. domestica [12,13,14,15,16].

Bornaviruses found in canaries are taxonomically grouped in the genus *Orthobornavirus* and form a separate species, which, according to a recent revision, has been named *Orthobornavirus serini*. At present, three viruses, Canary bornavirus 1, 2, and 3, are differentiated based on genetic features [17,18,19].

In contrast to the well-known role of parrot bornaviruses as disease agents in psittacines, the significance of canary bornaviruses as pathogens in canaries is still largely unclear. In three experimental infection studies using canary bornavirus isolates CnBV-1 and CnBV-2 obtained by cell cultivation, canaries were successfully infected as demonstrated by viral shedding and the detection of specific antibodies. The canaries, however, did not develop signs of disease during an observation period of up to five months [5,12,20]. 

There are contrastive reports of natural bornavirus infections in canaries which include, at least in a portion of the infected birds, the occurrence of clinical symptoms very similar to PDD in parrots. At present, these reports are limited to a few cases among individual birds kept in Austria, Germany, Poland, Italy, and Brazil [12,13,14,15,21,22]. The case report from Austria represented the first description of canary bornaviruses and referred to a canary that had died after a three-day disease with apathy and a dilated proventriculus. Bornavirus RNA and a non-suppurative inflammation in neuronal and non-neuronal tissues were detected, which are changes known from PDD in psittacines [13]. This first report was followed by a description of natural infections in living canaries in 12 out of 30 flocks in Germany leading to the conclusion that, at least in Germany, bornaviruses are common in canaries. Clinical signs, however, were only briefly described for three flocks in that study and included diarrhea, dilated proventriculus, or neurological signs [12]. In Poland, bornavirus RNA was detected in two flocks and a single canary each. One of these birds showed central nervous signs while the other bird revealed a dilated proventriculus during necropsy [14]. The occurrence of canary bornavirus in a canary in Italy has been documented so far only by a genomic sequence deposited in GenBank (accession number JQ08366). In Brazil, bornaviruses were found in two flocks, in one canary each. Sudden death was documented for one of the birds, while the other one showed apathy, anorexia, and a dilated proventriculus before death [16].

The case reports mentioned above might suggest the occurrence of a PDD-like disease in bornavirus-infected canaries, but they cannot be evaluated as evidence for a causal relation or correlation of bornavirus infection with the known non-pathognomonic PDD-like gastrointestinal and central nervous signs. This means that the signs might also be caused by other infectious or non-infectious agents. Consequently, there is, in contrast to the situation in psittacines, still a need to clarify the role of canary bornaviruses as a cause of disease. The additional question arises of whether canary bornaviruses in canaries, as distinct from parrot bornaviruses in psittacines, might represent facultative pathogens that need promotive factors to cause visible disease. These factors might be diverse and include co-infections with other viruses. Circoviruses especially, but also polyomaviruses which frequently infect canaries, might be relevant. Both viruses have been regarded as a cause of immunosuppression in birds and have been frequently detected together with other infectious agents [15,23,24].

Using an epidemiological approach, we aimed at answering the following questions. First: Do natural infections with canary bornaviruses lead to an increased frequency of PDD-like disease with gastrointestinal and central nervous signs in canaries? Second: Do concurrent infections with other viruses (circoviruses, polyomaviruses, adenoviruses, herpesviruses), yeasts, or trichomonads favor the occurrence of PDD-like signs in canary bornavirus-infected canaries? This study reveals that PDD-like signs were associated with a positive bornavirus test result in a collective of canary patients. Our data revealed that circovirus infections increased the chance of the appearance of PDD-like signs, but the effect was less distinct than that of bornaviruses. However, the data did not reveal that a simultaneous detection of both viruses reinforced the occurrence of PDD-like signs.

## 2. Materials and Methods

### 2.1. Ethical Statement

The study was performed according to institutional and national standards for the care and use of animals. It was approved by the Ethics Committee of the Veterinary Faculty, Ludwig-Maximilians-Universitaet Muenchen (Az. 305-21-04-2022). 

### 2.2. Study Design

A retrospective cohort study was performed using data and information about common canaries covering a period from 2009 (the start of bornavirus testing) until the end of June 2022, available at the patient data bank of a bird clinic located in Southern Germany. The investigation included canaries for which the case history or any clinical signs were documented, and which had been tested for the presence of bornavirus RNA by reverse transcription PCR offered by the internal diagnostic laboratory of the clinic. The clinical history including a description of disease signs was provided by the owners, the referring veterinarians or was recorded during a visit or a stay of one or more days duration in the bird hospital. Personal data of the owners were anonymized to be solely used for an affiliation of the birds to individual flocks or holdings. The investigations were requested independently of the present study and were based on medical indications or the wish of the owners. Consequently, not all examinations had been performed on every bird. The clinical samples of the birds originated from three sources: ambulatory or hospitalized patients of the clinic, birds for which clinical samples had been sent in from outside with a request for virus investigations offered by the diagnostic services of the clinic, or originating from dead birds sent to the necropsy laboratory to determine the cause of death.

Data from canaries with detected natural bornavirus infection were compared with those from canaries for which a negative test result was documented. If available, results of diagnostic tests for other viral, fungal, or parasitic co-infections were included in the comparison. The investigation presented here focused on co-infecting circovirus and polyomavirus which are particularly tested in canaries but also included adenovirus and herpesvirus, *Macrorhabdus ornithogaster* and other yeasts as well as trichomonads.

### 2.3. Diagnostic Tests

Clinical samples originating from live animals included swabs taken from choana, crop, and/or cloaca, while samples taken from dead birds mostly consisted of pools of varying organs such as the brain, liver, spleen, kidneys, and gastrointestinal tract. In several cases, birds were included with nucleic acid solutions being available as residual materials from other investigations which were used for additional bornavirus diagnostics.

RNA and DNA extraction from swabs and tissues was performed using the RNeasy mini kit and, additionally, the DNeasy blood and tissue mini kit (Qiagen, Hilden, Germany) or using the Indispin Pathogen Kit (Indical, Leipzig, Germany) according to the instructions of the manufacturers. Reverse transcription using random primers according to standard procedures was followed by PCR amplifying fragments of the bornavirus N or M gene [2,12]. In a varying number of birds, broad range circovirus [25], canary and finch polyomavirus (*Gammapolyomavirus secanaria* and *Gammapolyomavirus pypyrrhula*) [26], broad range adenovirus [27] and broad range herpesvirus [28] PCR tests were performed using the primers shown in Table 1, and as described before. PCR was analyzed by agarose gel electrophoresis, and products of expected sizes were purified using the QIAquick gel extraction kit (Qiagen, Hilden, Germany) and Sanger-sequenced by a commercial company (GATC Eurofins Genomics, Ebersberg, Germany) using the respective PCR primers. The identity of the PCR products was confirmed by BLAST analyses (https://blast.ncbi.nlm.nih.gov/, accessed at the investigation dates between 2009 and June 2022).

Microscopic investigations for *Macrorhabdus ornithogaster* or other yeasts as well as for trichomonads were based on direct native smears of crop swabs or of fecal samples under 400-fold magnification.

### 2.4. Canary Bornavirus Sequence Analyses

Partial N gene sequences obtained from 31 birds by the Kistler PCR protocol most used in this investigation [2], available in GenBank under the accession numbers OP150416-OP150446, were phylogenetically analyzed to verify the present taxonomic concept of the existence of a single species *Orthobornavirus serini* with three viruses. Sequences from PCR products of the other two PCR protocols used here were not included because of a lack of sequence overlap. In addition to the sequences obtained in this investigation (excluding the primer regions), representative sequences of the three canary bornaviruses as well as representative sequences of the other eight orthobornavirus species currently accepted were included. These sequences were obtained from GenBank (www.ncbi.nlm.nih.gov/nucleotide, accessed on 29 July 2022). Sequence alignments were performed using CLUSTALW included in Mega 11. The Kimura 3-parameter with a discrete Gamma distribution with 5 rate categories was determined as the best-fitting substitution model using Mega 11 [29,30] and was used in a neighbor-joining analysis to infer phylogenetic relations. Support for phylogenies was measured by bootstrapping 1000 replicates [31].

### 2.5. Analysis of Patient Data

For each canary included in the investigation, the patient database was evaluated for the results of the bornavirus diagnostic PCR, and in positive cases, the virus type was recorded. In addition, documented clinical history and clinical signs were analyzed regarding the presence or absence of symptoms characteristic of PDD. These included gastrointestinal signs (diarrhea, regurgitation, dilated proventriculus, secretion of undigested seeds with the feces), central nervous signs (coordination problems, seizures, tremors), or ocular disorders (impaired vision, chorioretinitis). Results of PCR investigations for circovirus, polyomavirus, adenovirus, and herpesvirus were included as detected, not detected, or not performed. Similarly, the results of microscopic investigations of direct native smears of swabs of crop and fecal samples for *Macrorhabdus ornithogaster* or other yeasts as well as for trichomonads were included in the analysis, if available, as detected, not detected, or not performed. 

Based on these nominal data, statistical analyses were performed by univariate analysis using chi-square statistics for the variables (the virus, fungus, or parasite tests performed using PDD-like signs as the dependent variable, each). When the expected case number was below 5, Fisher’s exact test was used to decide on independence. Odds ratios and *p* values were calculated to measure the strength of the relationship with *p* < 0.05 in the chi-square test and *p* < 0.1 in Fisher’s exact test regarded as significant. Cramér’s V values (phi values) were determined as a measure of effect size, and phi = 0.1 was interpreted as a weak effect, 0.3 as a medium effect, and 0.5 as a strong effect. In addition, multivariate linear regression modeling was used as a second method to determine a possible effect of the sample source, as well as to determine the association of bornavirus and circovirus test results with the occurrence of PDD-like signs, and to search for possible interaction effects of both viruses. All statistical analyses were performed using IBM SPSS Statistics version 28.0 (IBM Corp, Armonk, NY, USA).

## 3. Results

### 3.1. Bornaviral RNA Detection in Canaries

#### 3.1.1. Bird Collective

A total number of 201 canaries was included in the analyses. Seventy-five birds were male and 86 were female. The sex was unknown for 40 birds. Information on the age of the birds was not available, apart from the fact that none of them was a nestling. A total of 102 birds had been patients of the bird clinic and had been sampled as live birds by swabs, while for five birds, swab samples had been referred to the virological lab by veterinarians. A total of 94 birds had been sampled by internal organs during necropsy. 

The 201 birds were kept by 140 different owners, or, in other words, in 140 flocks. Flocks with six or more canaries, here named large flocks, contributed 106 birds to the investigation, while flocks consisting of 1–5 canaries, here regarded as small flocks, delivered 73 birds. For 22 canaries, the size of the flock was unknown.

#### 3.1.2. Bornavirus Diagnostics Results and Genetic Virus Characterization

Canary-bornavirus RNA was detected in 40 out of the 201 canaries (19.9%). These included 13 out of 75 males (17.3%), 25 of the 86 females (29.1%), as well as two of 40 canaries with unknown sex (5.0%). While swabs of 12 live patients of the clinic were PCR-positive (corresponding to 11.8%), RNA of bornavirus was found in the organs of 27 birds sampled during necropsy (28.7%), and in swabs of one out of the five birds (20%) submitted for virus diagnostics. When the detection rate in swab samples from bird clinic patients was compared with the detection rate in organ pools from necropsy cases, chi-square test resulted in *X*^2^ (1, *n* = 196) = 8.827, *p* = 0.003, φ = 0.212. The kind of sample (organ pools or swabs) was not independent of the rate of bornavirus RNA detection. Bornavirus RNA was detected in a higher portion of organ pools than in swabs taken from live clinic patients.

The canary-bornavirus-positive canaries originated from a total of 28 out of 140 flocks (20%), and most flocks delivered a single positive bird. Twenty-seven infected canaries originated from a total of 17 large flocks. Thirteen flocks contributed one positive bird each, while the remaining flocks delivered more than one positive bird (two flocks with two infected birds each, one flock with three birds, and one flock with seven birds). Ten small flocks contributed a total of eleven bornavirus-positive birds.

Based on BLAST analyses of the PCR products, CnBV-1 was identified in 21 birds originating from 16 flocks (owners), while CnBV-2 was found in eight birds from eight flocks and CnBV-3 was detected in eleven birds originating from six different flocks. The presence of more than a single virus type was detected only in those two flocks which had delivered three and seven positive birds. Here, CnBV-1 and CnBV-3 were found.

Besides BLAST analyses, phylogenetic analyses performed using sequences obtained from a partial N gene PCR [2] supported the assignment of the sequences to the three canary bornaviruses known so far (Figure 1). Two virus types each were detected in two flocks suggesting at least two entry events (Figure 1).

### 3.2. Clinical Signs

Clinical PDD-like signs similar to those known from diseased parrots were summarized as follows: Occurrences of diarrhea, regurgitation, dilated proventriculus, and excretion of undigested seeds were summarized as gastrointestinal (GIT) signs. Ataxia, seizures, and tremors were summarized as central nervous system (CNS) signs. Ocular disorders (impaired vision caused by chorioretinitis) were searched and detected only in a few birds. These signs suggest disorders of the neuronal retina occurred always in combination with CNS signs. They were thus combined with the CNS signs for evaluation.

As shown in Table 2, signs restricted to CNS disorders occurred in six canary-bornavirus-positive birds and in 15 virus-negative birds, while GIT signs only were reported for 14 bornavirus-positive and 17 bornavirus-negative birds. A combination of CNS and GIT signs was documented for eight bornavirus-positive and three bornavirus-negative birds. The presence of CNS signs only, GIT signs only, CNS-GIT in combination, or the presence of PDD-like signs (at least one of the signs mentioned before) did not depend on the virus type (in all tests *p* > 0.5). In total, the presence of PDD-like signs was recorded for 28 of 40 canary bornavirus-positive birds (70.0%) and for 35 of 161 bornavirus-negative birds (21.7%), while the absence of PDD-like signs was documented for 30% of the bornavirus-positive canaries and for 78.3% of the bornavirus-negative canaries of this collective. 

A chi-square test revealed that there was a significant lack of independence between bornavirus detection and the occurrence of PDD-like signs, *X*^2^ (1, *n* = 201) = 34.68, *p* < 0.001, φ = 0.415 (Table 3 and Table 4). The odds ratio was 8.4 (95% CI 3.9–18.2). PDD-like signs were more frequent in bornavirus-positive canaries than in bornavirus-negative canaries with a medium effect size. The null hypothesis that detection of bornavirus RNA and occurrence of PDD-like signs are independent, had to be rejected.

Separate analyses of data from live clinic patients and from birds sampled during necropsy confirmed these results with *X*^2^ (1, *n* = 102) = 13,61, *p* < 0.001, φ = 0.365, and an odds ratio of 9.9 (95% CI 2.4–39.8) for the live patients and *X*^2^ (1, *n* = 94) = 20.68, *p* < 0.001, φ = 0.469, and an odds ratio of 9.0 (95% CI 3.3–24.8) for the birds sampled during necropsy. Multivariate linear regression analyses revealed that the bornavirus effect remained stable when controlling for the sample source. There were indeed significant differences (0.17; t = 2.934, *p <* 0.005) in detection rates between the samples but the sample source itself did not prove a significant influence on PDD. The bornavirus effect remained stable in magnitude and significance (Appendix A).

### 3.3. Viral Co-Infections

Circovirus DNA was detected in 27 out of 125 canaries (21.6%) of the collective tested for circovirus. Seven of the circovirus-positive birds were also bornavirus-positive. As shown in Table 5, all seven birds revealed PDD-like signs. Circovirus-positive canaries were more likely to show PDD-like signs than circovirus-negative birds in the bornavirus-positive birds based on the results of Fisher’s exact test (*p* = 0.062, odds ratio 1.75 (95% CI 1.21–2.54) as well as in all birds tested for circovirus using the chi-square test with *X*^2^ (1, *n* = 130) = 4.354, *p* = 0.037, with φ = 0.183 suggesting a weak effect size (Table 4). The null hypothesis that the detection of circovirus DNA and the occurrence of PDD-like signs are independent in the group of bornavirus-positive canaries and in all canaries tested, had to be rejected. When the subgroup of bornavirus-negative canaries tested for circovirus was analyzed, circovirus infection status was shown to be independent of the occurrence of PDD-like signs with *p* = 0.365 obtained in Fisher’s exact test (Table 4). Circovirus co-infection status was independent of the bornavirus virus type (CnBV-1, CnBV-2 or CnBV-3, *p* = 0.278).

Multivariate logistic regression (Table 6) confirmed that circovirus positive tests increased the chance for PDD-like signs independently of the bornavirus test status. When controlling for the circovirus status, being bornavirus-positive increased the odds of showing PDD-like signs (model 2). An interaction model (model 3) did not reveal a joint interaction effect (Table 6) meaning a positive test status for both virus types did not reinforce the occurrence of PDD-like signs. Due to the low number of cases, the results have to be interpreted carefully, but linear probability models point in the same direction.

Polyomavirus DNA was detected in 28 out of 121 canaries tested by PCR for polyomavirus (23.1%). In ten birds, both bornavirus RNA and polyomavirus DNA was found (Table 7). As shown in Table 4, polyomavirus DNA detection status and the occurrence of PDD-like signs were shown to be independent. Polyomavirus co-infection status was independent of the bornavirus virus type (CnBV-1, CnBV-2 or CnBV-3, *p* = 0.619).

The adenovirus detection rate was low in the canaries included in this investigation. Only three of 83 birds included in the Adenovirus PCR test revealed a positive result. These three birds tested negative in the bornavirus assay, and one of the three birds revealed PDD-like signs. Herpesvirus DNA was not detected in 58 canaries investigated.

### 3.4. Co-Infections with Fungi and Trichomonads

*Macrorhabdus ornithogaster* was microscopically detected in five of 31 bornavirus-positive and 28 of 119 bornavirus-negative canaries (Appendix A). The incidence of PDD-like signs was independent of the *Macrophabdus* test status (Table 4). Independence was also demonstrated when GIT signs only were regarded (odds ratio 1.1 (95% CI 0.4–3.1), *p* = 1.000). *Macrorhabdus* co-infection status was independent of the bornavirus virus type (CnBV-1, CnBV-2 or CnBV-3, *p* = 0.161).

Other yeasts, also investigated by microscopic investigations, were detected in three of 28 bornavirus-positive canaries and in 30 of 117 bornavirus-negative canaries (Appendix A). The detection of yeasts was independent of the occurrence of PDD-like signs (Table 4). Independence was also demonstrated when GIT signs only were regarded (odds ratio 0.8 (95% CI 0.2–2.5), *p* = 0.784). Co-infection with other yeasts was independent of the bornavirus virus type (CnBV-1, CnBV-2, or CnBV-3, *p* = 0.393).

Trichomonads were found by microscopic investigations in 3 of 18 bornavirus-positive birds and in 25 of 89 bornavirus-negative birds (Appendix A). The occurrence of these parasites was independent of the occurrence of PDD-like signs (Table 4). Independence was also demonstrated when GIT signs only were regarded (odds ratio 0.4 (95% CI 0.1–2.1), *p* = 0.348). Co-infection with trichomonads was independent of the bornavirus virus type (CnBV-1, CnBV-2 or CnBV-3, *p* = 0.278).

## 4. Discussion

The relevance of canary bornaviruses as pathogens in common canaries has been unresolved so far. A lack of clinical disease symptoms during observation periods of up to five months in experimental infections of canaries [5,12,20] was in contrast to a few case reports of a PDD-like disease in natural infections of single birds [12,13,14,15]. The investigation presented here, based on a large number of canary patients of a bird clinic, could reveal for the first time, that infections with canary bornavirus are associated with PDD-like disease symptoms in common canaries. The results of the chi-square test for independence with *X*^2^ (1, *n* = 201) = 34.68, *p* < 0.001, φ = 0.415, clearly lead to the rejection of the null hypothesis that both parameters are independent. It could be concluded that PDD-like signs were more frequent in bornavirus-positive canaries than in bornavirus-negative canaries with a medium effect size. An odds ratio of 8.4 (95% CI 3.9–18.2) supported this conclusion. GIT and CNS signs, which are not pathognomonic for PDD or PDD-like diseases and might be the result of a variety of other causes, were also reported for the group of bornavirus-negative canaries, but with a much lower frequency. 

Reasons for the discrepancies between the previous experimental investigations and the natural infections in common canaries presented here are speculative. Many factors with potential influence on the course of natural infections elute evaluation. For example, the exact timepoint of natural infection is unknown for the canaries of this study, but the duration of infection might be an important factor in the appearance of the clinical signs of disease. PDD in parrots is known to have a varying, but mostly very long incubation time, sometimes even years, and to have a chronic course. It might be speculated that the natural infections of the birds in this investigation had persisted longer than in the experimental infections which were restricted to observation periods of up to five months. This experimental timeframe might have been too short to identify the signs of disease or even distinct inflammatory changes, which are constantly detected in bornavirus-infected psittacine birds. Additionally, it might be difficult to recognize the early stages of disease in canaries which are known to hide signs of disease, a phenomenon typical of birds living in flocks and as prey in the wild [32]. Age might be a very important factor for infection or the development of clinical symptoms, but, because of missing data, cannot be evaluated here. Other factors such as breed and individual disposition of the birds, as well as the keeping or feeding conditions, might have influenced the clinical situation as well.

Regarding the investigation presented here, further possible limitations and resulting bias need to be discussed. The sample of canaries included here did not arise from a randomized trial study, and therefore the selection criteria cannot be precisely defined. The descriptions of clinical signs were from various sources. They partly came from bird owners, and though many of them observe their birds very carefully, their medical knowledge is unknown. However, it seems unlikely that available records or thoroughness of clinical records may have differed between bornavirus-positive and -negative individuals because bornavirus diagnostics were generally performed after recording the history. Histology which might have indicated subclinical pathological disorders regarded as typical for PDD in psittacine birds was not included in the analysis. 

All bornavirus genomes detected in canaries unequivocally grouped with one of the three canary bornaviruses known so far, CnBV-1, CnBV-2, or Cn-BV-3, as shown for those cases where partial N gene fragments produced by a protocol according to Kistler et al. [2] was obtained and could be included in the phylogenetical analysis. This affiliation to one of the three canary bornaviruses known so far was, however, also true for PCR products obtained by a PCR targeting another N-gene fragment [12] or an M gene PCR [2] which were evaluated by BLAST analyses only. Our investigation thus confirmed the present taxonomic concept of canary bornaviruses. 

The second scientific question addressed in this investigation was, whether viral, fungal, or parasitic co-infection influences the occurrence of PDD-like signs in bornavirus-infected canaries. Circoviruses, for which test results were available for 125 canaries, were shown to have an effect on the occurrence of PDD-like signs. This effect was visible in the group of bornavirus-positive canaries with *p* = 0.062, obtained in Fisher’s exact test and an odds ratio of 1.75 (95% CI 1.21–2.54), but to our surprise, also in the whole group of circovirus-tested canaries (*X*^2^ (1, *n* = 130) = 4.354, *p =* 0.037, φ = 0.183 and an odds ratio of 2.49). The effect size of 0.183, estimated as weak, was much smaller than that of bornavirus assessed as medium-sized (φ = 0.415). It is not known how canary circoviruses influence the course of the PDD-like disease. Avian circoviruses are generally regarded as having an immunosuppressive effect [23]. Atrophy of lymphatic organs, especially of the Bursa of Fabricius [33], and increased B lymphocyte apoptosis [34] have been shown in pigeons infected with pigeon circovirus. We were thus expecting an interaction effect with circoviruses enhancing the pathogenic effect of bornaviruses. Such an interaction effect could, however, not be proven using multivariate logistic regression. A possible reason might be the rather small case number, especially in the group of bornavirus-positive canaries co-infected with circoviruses. Furthermore, we cannot rule out a selection bias because the birds of this study were patients of a bird clinic and tested for circovirus and bornavirus because of partly unknown reasons. Future studies including more birds or focusing on pathogenicity mechanisms might answer this question. 

Polyomaviruses, in contrast to canary circoviruses, did not show any association with the occurrence of PDD-like signs. This was noted in the bornavirus-positive canaries as well as in the bornavirus-negative canaries and in all canaries tested for this virus.

It has to be noted that the detection rates of circoviruses and polyomaviruses in the collective of canaries of this investigation reached 21.6% and 23.1%, respectively, representing remarkably high values that are very similar to that of bornavirus (19.9%) suggesting high importance and a need for more research on these viruses in common canaries. 

Associations of adenovirus or herpesvirus infections with PDD-like signs were not evaluated because of the low number of birds testing positive for those viruses. Surprisingly, associations of other pathogens including *Macrorhabdus ornithogaster*, other yeasts, or trichomonads with gastrointestinal or central nervous system signs were not detected, either in the bornavirus-positive, the bornavirus-negative or all canaries tested. These pathogens are generally regarded as important differential diagnoses for disorders of the gastrointestinal tract or the central nervous system in birds [35].

In the investigation presented here, canary bornaviruses were detected by commonly used RT PCR assays [2,12] and using swabs taken from choana, crop, and cloaca as well as pools of internal organs (such as brain, gastrointestinal tract, liver, kidneys). It is remarkable that the detection rate of 11.9% in the swab samples taken from 101 live bird patients of the clinic was lower than that of 28.7% based on testing of organs taken from 94 canaries during necropsy. Association of bornavirus detection and PDD-like signs were detected in both bird groups independently, even though the detection rate of bornavirus differed between the swab and tissue samplings. Most likely, this discrepancy reflects a lower virus content in swabs compared to internal organs and indicates a probable underestimation of positivity rates in the live canary patients included in this investigation or even in general in live canaries tested for this pathogen. During experimental bornavirus infections of canaries, differences in virus distributions were detected at the end of the five-month observation period. While some birds revealed continuous shedding of viral RNA, other birds were negative for shedding as shown by negative swab samples, but high viral loads were detected in their brains [5]. These results emphasize a need for improving bornavirus diagnostics in live canaries. In parrots, antibody detection tests using immunofluorescence assays or ELISA improved the identification of infected live psittacine birds and are commonly used [36], (own observation). In canaries, however, serological tests for detecting specific anti-bornavirus antibodies have not been established yet as a routine diagnostic tool. Serology has therefore not been used in the investigation presented here which was performed in a clinical setup. In canaries, blood sampling does not generally represent a routine diagnostic procedure because of the risk of high accidental blood losses and a generally high risk of blood sampling in sick and weakened light-weighted small fringillid finches, leading at present, to limitations in bornavirus detection in canaries. 

It has to be noted that the canary bornavirus detection rate of about 20% obtained for the birds in this investigation does not represent true prevalence data since the birds were not part of a representative randomized study but were patients of a bird clinic and were thus preselected, although the selection criteria cannot be precisely defined. Detection in 28 out of 140 flocks corresponding to a detection rate of 20% is somewhat lower than that of an earlier study also performed in Germany, where canary bornaviruses had been detected in 12 of 30 flocks (40%) [12]. Our study nevertheless confirmed canary bornaviruses as a frequent infection in common canaries, at least in Germany.

In conclusion, clinical data including a large number of cases presented here allowed, for the first time, scientific conclusions about the importance of canary bornaviruses as pathogens. They revealed that canary bornaviruses are associated with a PDD-like disease in common canaries, similar to parrot bornaviruses in psittacine birds. In addition, our data suggest circoviruses as a factor supporting the occurrence of PDD-like signs.

## Figures and Tables

**Figure 1 viruses-14-02187-f001:**
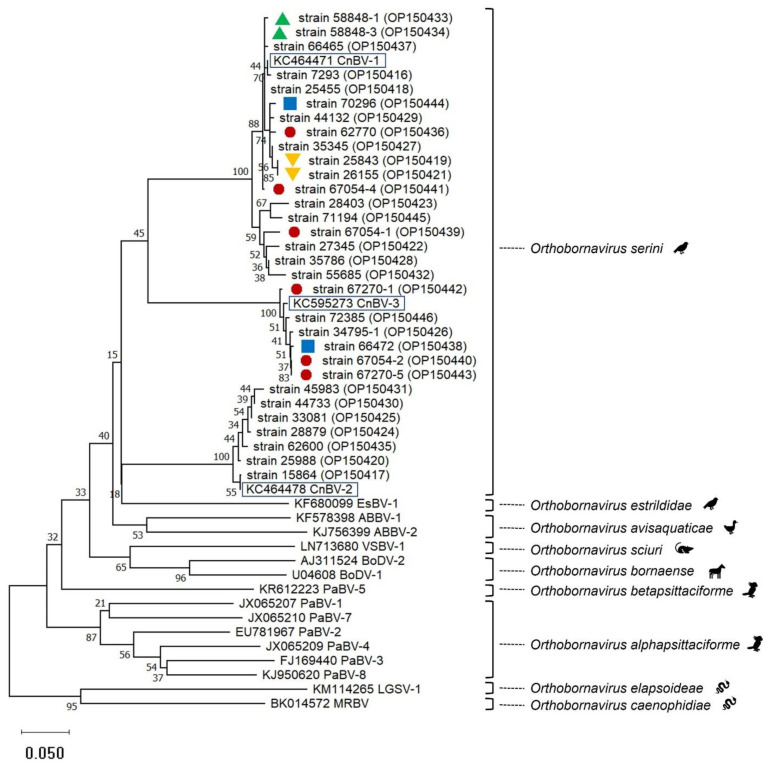
Phylogenetic relationship based on CLUSTALW alignments and neighbor-joining analyses including bornavirus partial N gene nucleotide sequences (0.35 bp) from 31 canaries (this study, strain names with GenBank accession numbers in brackets are given) and representatives of the 9 species included in the genus *Orthobornavirus* (GenBank accession numbers starting with letters are given). Representatives of the three viruses included in the species *Orthobornavirus serini* are highlighted by black boxes. Red circles, blue squares, as well as green or orange triangles, mark origin from the same flocks. The other canary-bornavirus strain sequences of this tree originated from different flocks. Sequences obtained in this investigation are available in GenBank under the accession numbers OP150416-OP150446.

**Table 1 viruses-14-02187-t001:** PCR-based methods to detect viruses in clinical samples from canaries.

Virus	Assay ^1^	Primers Used	Refs.
Bornavirus	RT PCR	ABV_NconsensusF/ABV_NconsensusR; ABV_MconsensusF/ABV_MconsensusR; Ccon_490/Ccon_708	[2,12]
Circovirus	Nested PCR	Cv-s/Cv-as, Cn-s/Cn-as	[25]
Polyomavirus	PCR	APV-C1/APV-C2	[26]
Adenovirus	Nested PCR	polFouter/polRouter, polFinner/polRinner	[27]
Herpesvirus	Nested PCR	DFA, ILK/KG1, TGV/IYG	[28]

^1^ RT-PCR, reverse transcription polymerase chain reaction, PCR, polymerase chain reaction.

**Table 2 viruses-14-02187-t002:** Occurrence of canary bornavirus types and PDD-like signs in 201 canaries.

Virus	Number of Birds	CNS Signs ^1^ Only	GIT Signs ^2^ Only	CNS and GIT Signs Combined	PDD ^3^-like Signs Absent
CnBV-1 positive	21	3	9	5	4
CnBV-2 positive	8	1	2	1	4
CnBV-3 positive	11	2	4	2	4
Total positive	40	6	14	8	12
Virus negative	161	15	17	3	126

^1^ CNS signs, central nervous system signs (coordination problems, seizures, tremors, chorioretinitis); ^2^ GIT signs, gastro-intestinal tract signs (diarrhea, regurgitation, dilated proventriculus, excretion of undigested seeds); ^3^ PDD, proventricular dilatation disease.

**Table 3 viruses-14-02187-t003:** Cross table for the number of canaries with or without proventricular dilatation disease (PDD)-like signs and bornavirus test results. *X*^2^ (1, *n* = 201) = 34.68, *p* < 0.001, φ = 0.415. Expected counts for perfectly independent variables are given in brackets.

Test Status	PDD-like Signs Present	PDD-like Signs Absent	Total
Bornavirus positive	28 (12.5)	12 (27.5)	40
Bornavirus negative	35 (50.5)	126 (110.5)	161
Total	63	138	201

**Table 4 viruses-14-02187-t004:** Univariate (chi-square) analysis of factors potentially associated with the occurrence of PDD-like signs. Significant results are shown in bold.

Factor	Bird Group	Odds Ratio	95% CI	*p*-Value
**Bornavirus**	**Total**	**8.40**	**3.9–18.2**	**<0.001 ^1^**
Circovirus	**Bornavirus positive**	**1.75**	**1.21–2.54**	**0.062 ^2^**
	Bornavirus negative	1.77	0.59–5.32	0.365^2^
	**Total**	**2.49**	**1.04–5.94**	**0.037 ^1^**
Polyomavirus	Bornavirus positive	1.60	0.23–11.08	1.000 ^2^
	Bornavirus negative	0.36	0.08–1.72	0.230 ^2^
	Total	1.15	0.47–2.79	0.820 ^1^
Macrorhabdus	Bornavirus positive	0.67	0.09–4.80	1.000 ^2^
	bornavirus negative	1.35	0.50–3.67	0.553 ^1^
	Total	0.98	0.42–2.27	0.959 ^1^
Other yeasts	Bornavirus positive	1.47	1.12–1.92	0.536 ^2^
	Bornavirus negative	1.25	0.46–3.40	0.651 ^1^
	Total	1.00	0.43–2.32	0.995 ^1^
Trichomonads	Bornavirus positive	1.00	0.07–13.87	1.000 ^2^
	Bornavirus negative	0.75	0.24–2.34	0.618 ^1^
	Total	0.70	0.26–1.80	0.436 ^1^

^1^ chi-square test; ^2^ Fisher’s exact test.

**Table 5 viruses-14-02187-t005:** Cross table for the number of canaries with positive or negative bornavirus tests as well as circovirus tests and occurrence of proventricular dilatation disease (PDD)-like signs. Expected counts for perfectly independent variables are given in brackets.

	Test Status	PDD-like Signs		Total
		Present	Absent	
Bornavirus positive ^1^	Circovirus positive	7 (4.8)	0 (2.3)	7
	Circovirus negative	12 (14.3)	9 (6.8)	21
	Total	19	19	28
Bornavirus negative ^2^	Circovirus positive	6 (4.3)	14 (15.7)	20
	Circovirus negative	16 (17.7)	66 (64.3)	82
	Total	22	80	102
All canaries ^3^	Circovirus positive	13 (8.5)	14 (18.5)	27
	Circovirus negative	28 (32.5)	75 (70.5)	103
	Total	41	89	130

^1^ Fisher’s exact test *p* = 0.062, odds ratio 1.75 (95% CI 1.21–2.54); ^2^ Fisher’s exact test *p* = 0.365; ^3^ Chi-square test *X*^2^ (1, *n* = 130) = 4.354, *p* = 0.037, φ = 0.183.

**Table 6 viruses-14-02187-t006:** Results of logistic regression analysis with occurrence of proventricular dilatation disease (PDD)-like signs as the dependent variable. Estimates were calculated using virus test status as independent variables (bornavirus in model 1, bornavirus and circovirus in model 2, bornavirus, circovirus, and interaction in model 3). *p*-values < 0.05 were regarded as significant.

Model	Variable	95% CI	Odds Ratio	*p*-Value
1 ^1^	(constant)		0.28	0.000
	bornavirus	3.05–19.318	7.68	0.000
2 ^2^	(constant)		0.22	0.000
	Bornavirus	3.06–20.14	7.85	0.000
	Circovirus	1.01–6.79	2.61	0.049
3 ^3^	(constant)		0.242	0.000
	Bornavirus	1.98–15.29	5.50	0.001
	Circovirus	0.59–5.32	1.77	0.311
	Interaction Borna-Circovirus	0−>100	>100	0.999

^1^ Cox-Snell R^2^ 0.146; ^2^ Cox-Snell R^2^ 0.171; ^3^ Cox-Snell R^2^ 0.194.

**Table 7 viruses-14-02187-t007:** Cross table for the number of canaries with positive or negative bornavirus tests, as well as polyomavirus tests, and occurrence of proventricular dilatation disease (PDD)-like signs. Expected counts for perfectly independent variables are given in brackets.

	Test Status	PDD-like Signs		Total
		Present	Absent	
Bornavirus positive	Polyomavirus positive	8 (7.5)	2 (2.5)	10
	Polyomavirus negative	10 (10.5)	4 (3.5)	14
	Total	18	6	24
Bornavirus negative	Polyomavirus positive	2 (4.1)	16 (13.9)	18
	Polyomavirus negative	20 (17.9)	58 (60.1)	78
	Total	22	74	86
All canaries	Polyomavirus positive	10 (9.3)	18 (18.7)	28
	Polyomavirus negative	30 (30.7)	62 (61.3)	92
	Total	40	80	120

## Data Availability

The bornaviral partial N gen sequences generated in this study were submitted to the GenBank database (https://www.ncbi.nlm.nih.gov/genbank/, accessed on 28 July 2022) under accession numbers OP150416-OP150446.

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
