# Peer review of "Canary Bornavirus (Orthobornavirus serini) Infections Are Associated with Clinical Symptoms in Common Canaries (Serinus canaria dom.)"

_viruses, 2022, doi:10.3390/v14102187_

Round 1

Reviewer 1 Report

Rinder et al. tested samples collected from domestic canaries for canary bornaviruses and other potential pathogens and analysed their association with reports on clinical signs that might indicate proventricular dilatation disease (PDD). They observe a statistically significant association between the detection of canary bornaviruses and the occurrence of clinical signs possibly indicating PDD. Based on this observation, they claim canary bornaviruses to be a primary cause of PDD-like signs.

The manuscript is well written, but the study appears to have limitations with regard to the available sample and data collection. Given the remarkable discrepancy between the results of this study and previous experimental studies for bornavirus infections of canaries, the possible limitations of this study should be given thorough consideration and the conclusion should be toned down considerably.

Major issues:

Three experimental infection studies of canaries with canary bornaviruses has been published so far (only two of them are cited in the manuscript; Olbert et al., 2016 is missing). Despite persistent infection, the experimental birds did not develop PDD-like disease in any of these studies and likewise clear inflammatory changes were not induced (a hallmark of PDD, that is consistently detectable even in asymptomatically bornavirus-infected psittacines). The absence not only of clinical disease but also of subclinical indicators should be emphasized more clearly in the discussion (e.g. in the discussion section when speculating that the absence of clinical signs might be due to the duration of the experiments of only 5 months, lines 337-338).

The authors interpretation that their results “verify canary bornaviruses as a primary cause of PDD-like disease” (lines 23-24) are in sharp contrast to this published data. While the authors provide several approaches to an explanation why the canary bornaviruses did not induce lesions or clinical signs in the published studies, they should challenge their own results with the same thoroughness and discuss potential limitations and biases, caused e.g. by the apparently heterogeneous composition of the sample collection and variable sources of clinical data.

One potential bias is already discussed by the authors, but not in the context of the association bornaviruses and PDD. The authors observe that detection rates were clearly lower for swabs as compared to tissue samples by a factor of ~2.5 (line 197-199) and assume that detection rates in live birds may have been underestimated (lines 392-393). The lower sensitivity of detection of bornavirus RNA in swabs has been described also in various other studies. It may be reasonable to assume that many swabs may have originated from live animals without or with only mild signs of disease, whereas tissue samples were available mainly for animals that had died from severe disease. This may have contributed to the observed association between virus and disease. The authors should analyse the different sample types independently to avoid any bias of their results.

Furthermore, they should discuss whether available records or thoroughness of clinical examination may have differed between bornavirus-positive and negative individuals.

The definition of PDD-like disease in this study is based solely on a broad range of clinical parameters, which are by no means pathognomonic for PDD. Bornaviruses induce diseases by immunopathogenesis, which is characterized by typical inflammatory lesions (encephalitis, ganglioneuritis). Diagnosis of PDD in psittacines is based on these microscopic lesions in combination with typical clinical signs and/or gross lesions. A similarly strict definition of PDD-like disease might have been beneficial also for this study. While histopathological data was presumably not available for most animals in this study, this limitation of the study design should at least be discussed.

Overall, the observed association should be interpreted more carefully and cannot claimed as evidence for a causative role as long as experimental data is clearly discrepant and the reasons for this discrepancy remain unknown. Please rephrase your statements accordingly (lines 23-24; 87-88; 417-418).

Minor issues:

Title: “Orthobornavirus serini” needs to be italicized.

lines 37-38: Please rephrase this statement. Avian bornaviruses are a genetically highly diverse and not monophyletic group of viruses, which can be found in a broad range of avian species. However, the natural host range of each of these viruses is presumably rather narrow.

line 41: “Orthobornavirus” has to be italicized.

line 45: Please rephrase: “… the well-known role of PARROT bornaviruses …”, “the significance OF CANARY BORNAVIRUSES as pathogens in canaries …”.

line 46-50: A third experimental study on CnBV in canaries is missing (Olbert et al., 2016).

lines 56, 65-67: I suggest removing the citation of the GenBank entry.

lines 76-77: Please rephrase: “… whether canary bornavirus IN CANARIES, as distinct from parrot bornaviruses IN PSITTACINES, …”.

Please acknowledge throughout the manuscript, that infections with the same bornavirus may have different outcomes in different host species. Thus, their characteristics may not always be generalized and the specification of virus and host is mandatory for many statements.

line 132: Instead of providing only viral families, please specify the range of viruses detected by each of the four PCR tests.

Table 1: Please correct typo: ABV_NconsensusR

line 146: Why were only 31 sequences generated from 40 positive individuals. Please explain how the samples were selected.

Figure 1: The phylogenetic tree is obviously displayed with the “transform to equal branches” option, which leads to equal branch lengths between neighbouring sequences regardless of their true genetic distances. This type of layout is highly misleading and has to be changed to a non-transformed format.

Please provide accession numbers of the newly generated sequences in the tree.

It would be highly informative to indicate sequences originating from the same flock directly in the tree. (Instead, the term “from canary” could be removed, since all new sequences originate from canaries.)

line 285: Typo: “shown”

line 379-385: Macrorhabdus, yeasts and trichomonads are known to be involved in intestinal disorders but usually not associated with neurologic signs. Thus, a definition of PDD-like signs that involves both may limit the chance of detecting an association for these pathogens. Are associations observed when only gastrointestinal signs are included?

Author Response

First of all, we would like to thank the reviewer for carefully reviewing the manuscript and for the very helpful remarks and suggestions to improve the manuscript. We have included the changes in the revised version of the manuscript as described below.

Three experimental infection studies of canaries with canary bornaviruses has been published so far (only two of them are cited in the manuscript; Olbert et al., 2016 is missing). Despite persistent infection, the experimental birds did not develop PDD-like disease in any of these studies and likewise clear inflammatory changes were not induced (a hallmark of PDD, that is consistently detectable even in asymptomatically bornavirus-infected psittacines). The absence not only of clinical disease but also of subclinical indicators should be emphasized more clearly in the discussion (e.g. in the discussion section when speculating that the absence of clinical signs might be due to the duration of the experiments of only 5 months, lines 337-338).

Thank you for mentioning the reference Olbert et al. (2016) which was in fact lacking. We now added the corresponding information and this reference. We also mentioned the absence of histologically detectable inflammatory changes in the experimental infections (lines 376-377).

The authors interpretation that their results “verify canary bornaviruses as a primary cause of PDD-like disease” (lines 23-24) are in sharp contrast to this published data. While the authors provide several approaches to an explanation why the canary bornaviruses did not induce lesions or clinical signs in the published studies, they should challenge their own results with the same thoroughness and discuss potential limitations and biases, caused e.g. by the apparently heterogeneous composition of the sample collection and variable sources of clinical data.

 Thank you for your remarks. We changed the word “verify” to “indicate” because our investigation did not evidence a causal relation, but an association. We further changed some words in lines 372-377 to make clear that we do not want to discuss the other studies (which was not our intention) but to point at differences which might explain the different results. We further added a section with remarks on the heterogenous sample and the variable sources of clinical data (lines 384-393).

One potential bias is already discussed by the authors, but not in the context of the association bornaviruses and PDD. The authors observe that detection rates were clearly lower for swabs as compared to tissue samples by a factor of ~2.5 (line 197-199) and assume that detection rates in live birds may have been underestimated (lines 392-393). The lower sensitivity of detection of bornavirus RNA in swabs has been described also in various other studies. It may be reasonable to assume that many swabs may have originated from live animals without or with only mild signs of disease, whereas tissue samples were available mainly for animals that had died from severe disease. This may have contributed to the observed association between virus and disease. The authors should analyse the different sample types independently to avoid any bias of their results.

We now performed additional statistical analyses to answer this question and added this information in the results (lines 273-276) and in the discussion (lines 442-444). In addition, we examined in a multivariate linear regression framework whether the bornavirus effect remains stable when controlling for the sample source. There are indeed significant difference in detection rates between the samples but the sample source itself does not have a significant influence on PDD and the bornavirus effect remains stable in magnitude and significance. Results are available upon request.

Furthermore, they should discuss whether available records or thoroughness of clinical examination may have differed between bornavirus-positive and negative individuals.

We added a comment on this in lines 389-391.

The definition of PDD-like disease in this study is based solely on a broad range of clinical parameters, which are by no means pathognomonic for PDD. Bornaviruses induce diseases by immunopathogenesis, which is characterized by typical inflammatory lesions (encephalitis, ganglioneuritis). Diagnosis of PDD in psittacines is based on these microscopic lesions in combination with typical clinical signs and/or gross lesions. A similarly strict definition of PDD-like disease might have been beneficial also for this study. While histopathological data was presumably not available for most animals in this study, this limitation of the study design should at least be discussed.

We now added a comment on this topic in lines 392-393.

Overall, the observed association should be interpreted more carefully and cannot claimed as evidence for a causative role as long as experimental data is clearly discrepant and the reasons for this discrepancy remain unknown. Please rephrase your statements accordingly (lines 23-24; 87-88; 417-418).

We agree that we investigated for associations which do not directly allow conclusions about causal relations. We therefore changed the wording in lines 23-24 and in line 472 (former 441) correspondingly.

Minor issues:

Title: “Orthobornavirus serini” needs to be italicized.

This was changed as suggested.

lines 37-38: Please rephrase this statement. Avian bornaviruses are a genetically highly diverse and not monophyletic group of viruses, which can be found in a broad range of 3 avian species. However, the natural host range of each of these viruses is presumably rather narrow.

We rephrased these sentences (see lines 37-41).

line 41: “Orthobornavirus” has to be italicized.

This was changed as suggested.

line 45: Please rephrase: “… the well-known role of PARROT bornaviruses …”, “the significance OF CANARY BORNAVIRUSES as pathogens in canaries …”.

Was changed as proposed (now line 46-47).

line 46-50: A third experimental study on CnBV in canaries is missing (Olbert et al., 2016).

Thank you for this remark. We now also mention this experiment and this reference.

lines 56, 65-67: I suggest removing the citation of the GenBank entry.

We would like to keep a reference for the detection of canary bornavirus in Italy but we removed the reference in line 65-67 (now line 69).

lines 76-77: Please rephrase: “… whether canary bornavirus IN CANARIES, as distinct from parrot bornaviruses IN PSITTACINES, …”.

Was changed as proposed (now line 79-80).

Please acknowledge throughout the manuscript, that infections with the same bornavirus may have different outcomes in different host species. Thus, their characteristics may not always be generalized and the specification of virus and host is mandatory for many statements.

Thank you for this advice. We now added the host names and the bornavirus species names in several cases where it might be unclear.

Line 132: Instead of providing only viral families, please specify the range of viruses detected by each of the four PCR tests.

We added information on the known specificities of the virus PCRs used in this investigation: PCRs for circovirus, adenovirus, and herpesvirus are broad range PCRs while the polyomavirus PCR is able to detect canary polyomaviruses and finch polyomaviruses (Gammapolyomavirus secanaria and Gammapolyomavirus pypyrrhula). Both polyomavirus species have been detected so far in domestic canaries (see lines 137-139).

Table 1: Please correct typo: ABV_NconsensusR

Was correct accordingly

line 146: Why were only 31 sequences generated from 40 positive individuals. Please explain how the samples were selected.

We know added further information to make clear why only 31 sequences were used in the tree (lines 155-157).

Figure 1: The phylogenetic tree is obviously displayed with the “transform to equal branches” option, which leads to equal branch lengths between neighbouring sequences regardless of their true genetic distances. This type of layout is highly misleading and has to be changed to a non-transformed format.

Thank you for this advice. We now replaced the figure with a tree without equal branches.

Please provide accession numbers of the newly generated sequences in the tree.

Was done as suggested.

It would be highly informative to indicate sequences originating from the same flock directly in the tree. (Instead, the term “from canary” could be removed, since all new sequences originate from canaries.)

We now removed the term “from canary” and instead marked the sequences with origins from the same flocks. We also added a corresponding note in the text (lines 229-230).

line 285: Typo: “shown”

Was corrected.

line 379-385: Macrorhabdus, yeasts and trichomonads are known to be involved in intestinal disorders but usually not associated with neurologic signs. Thus, a definition of PDD-like signs that involves both may limit the chance of detecting an association for these pathogens. Are associations observed when only gastrointestinal signs are included?

We now also examined this potential association with gastrointestinal signs only (lines 332-334, 339-340, 344-345).

All authors would like to thank the reviewers again for their time and support, and we hope that we could revise the manuscript to your full satisfaction.

Monika Rinder

Reviewer 2 Report

This is a very nice study of boraviruses in canaries. It supports and somewhat expands on earlier studies. The most interesting facet of the work is the attempt to determine if co-infections might exacerbate the development of PDD in bornavirus infected birds. The results presented here may lead to interesting follow-up studies.  

There are some minor issues with the writing and a few minor typos, as listed below. 

Lines 37 to 39.  I suggest simplifying and clarifying the sentence to read “The host range of avian bornaviruses is not restricted to psittacines but includes water fowl, estrildid finches and common or domestic canaries, Serinus canaria form. domestica.

Line 51. I suggest rewording the beginning of this sentence to:  “There are contrasting reports of natural bornavirus infections in canaries…..

Line 53. I suggest rewriting the sentence  that begins “At present..’ Perhaps: “At present these reports are limited to a few cases among individual birds kept in Austria….”

Line 87: Change to: This study reveals that PDD-like signs were associated with a positive bornavirus test results in a collective of canary patients.

Line 199: Change (28,7%) to (28.7%)

Line 277: Add a space before (model 2)

Line 309: Change ‘microscopical’ to ‘microscopic’

Line 336: Change ‘chronicle’ to ‘chronic’ 

Line 344 Change: ‘Bornavirus’ to ‘bornavirus’

Author Response

First of all we would like to thank the reviewer for carefully reviewing the manuscript and for the very helpful remarks and suggestions to improve the manuscript. We have included the changes in the revised version of the manuscript as described below.

Lines 37 to 39. I suggest simplifying and clarifying the sentence to read “The host range of avian bornaviruses is not restricted to psittacines but includes water fowl, estrildid finches and common or domestic canaries, Serinus canaria form. domestica.

The changes were included as suggested but with modification according to suggestions of reviewer 1.

Line 51. I suggest rewording the beginning of this sentence to:  “There are contrasting reports of natural bornavirus infections in canaries…..

The changes were included as suggested.

Line 53. I suggest rewriting the sentence  that begins “At present..’ Perhaps: “At present these reports are limited to a few cases among individual birds kept in Austria….”

The changes were included as suggested.

Line 87: Change to: This study reveals that PDD-like signs were associated with a positive bornavirus test results in a collective of canary patients.

The changes were included as suggested.

Line 199: Change (28,7%) to (28.7%)

Was done.

Line 277: Add a space before (model 2)

Was done.

Line 309: Change ‘microscopical’ to ‘microscopic’

Was done.

Line 336: Change ‘chronicle’ to ‘chronic’ 

Was corrected as suggested.

Line 344 Change: ‘Bornavirus’ to ‘bornavirus’

Was corrected as suggested.

All authors would like to thank the reviewer again for their time and support, and we hope that we could revise the manuscript to your full satisfaction.

Monika Rinder

Reviewer 3 Report

In this manuscript, to determine whether canary bornaviruses are associated with a PDD-like disease in common canaries, the authors retrospectively examined the clinical data of 201 canaries in a bird clinic in Southern Germany, where RT-PCR tests were conducted on bornaviruses as well as co-infectious agents such as circoviruses and polyomaviruses. The authors showed that canary bornavirus (CnBV) RNAs were detected in clinical samples of 40 out of 201 canaries. Furthermore interestingly, PDD-like signs were found associated with CnBV infection with circoviruses detection. This study demonstrated CnBVs cause a PDD-like disease in naturally infected canaries and also suggested that circovirus co-infection may have effect to promote the PDD-like syndrome in infected birds. This paper provides valuable information regarding the previously unclear association between CnBV infection and PDD-like syndrome, as well as the possibility of co-infection with other infectious agents as a factor promoting the development of PDD-like diseases. This reviewer requests responses to the following several points.

1. All of CnBV-1, -2, -3 have been detected in birds showing PDD-like symptoms. Are there any differences in PDD symptoms, age, or co-infection status with circoviruses or other infectious agents among birds with different CnBV genotypes?

2. The data showed that CnBV appears to be widespread among common canaries. Based on the phylogenetic data of detected CnBVs, it would be interesting to analyze or discuss when these viruses began to spread in Germany and via what route.

3. Also include an analysis of the age distribution of canaries with PDD-like diseases if possible.

Author Response

First of all we would like to thank the reviewer for carefully reviewing the manuscript and for the very helpful remarks and suggestions to improve the manuscript. We have included the changes in the revised version of the manuscript as described below.

  1. All of CnBV-1, -2, -3 have been detected in birds showing PDD-like symptoms. Are there any differences in PDD symptoms, age, or co-infection status with circoviruses or other infectious agents among birds with different CnBV genotypes?

We did not detect any differences in PDD symptoms or co-infection status with circoviruses, polyomaviruses, Macrorhabdus, other yeasts or trichomonads among birds with different CnBV genotypes (p >0.161 for all factors). We now added the additional information in the text (lines 297-298, 310-312, 334-335, 340-341, 346-3347).

We agree very much with the reviewer that age might be a very important factor influencing infection status or the development of clinical symptoms. We, however, do not have any information on the age of most of the birds except that we did not have any nestlings in the patient collective, so we cannot take any conclusions on the relevance of age. We now added this information in the results and included a comment in the discussion (lines 195-196, 380-382).

  1. The data showed that CnBV appears to be widespread among common canaries. Based on the phylogenetic data of detected CnBVs, it would be interesting to analyze or discuss when these viruses began to spread in Germany and via what route.

We agree that time and route of spreading is very interesting. To our opinion, the available data, however, do not allow any conclusions. Breeding of canaries is a hobby distributed in many countries of Europe and even in other continents, and canaries are extensively traded and exchanged between keepers. In most cases the owners of the canaries do not have any information on the breeding history or geographical origin of the parents or ancestors of the birds of their flocks. Conclusions about the origins and spreading of the viruses thus appear to be very difficult. To our opinion we even cannot speculate about this topic.

We now added some information about the flocks of origin to the tree. In two flocks with two or more positive birds included, two different virus types were detected suggesting at least two entry events. We now added this information in the text (lines 229-230). These aspects were not the topic of our paper, and we therefore did not go into further details.

  1. Also include an analysis of the age distribution of canaries with PDD-like diseases if possible.

As mentioned above, we agree that age at the time of infection might be an important aspect. We, however, do not have any information on the age of most of the birds except that we did not have any nestlings in the bird collective, so we cannot take any conclusions about the relevance of age. We now added this information in the results and included a comment in the discussion (lines 195-196, 379-381).

All authors would like to thank the reviewer again for the time and support, and we hope that we could revise the manuscript to your full satisfaction.

Monika Rinder

Round 2

Reviewer 1 Report

The authors addressed all relevant issues and considerably improved the manuscript. I have only two issues left:

1. Since the authors performed a multivariate linear regression analysis supporting their hypothesis, they should include it also in the manuscript and not only make its results available on request.

2. I suggest to phrase the statement in lines 23 to 25 more carefully by using terms such as “to be involved” or “to contribute to” instead of “to be a cause”.

Author Response

Thank you again for carefully reviewing the manuscript. We now modified the manuscript as listed above:

  1. Since the authors performed a multivariate linear regression analysis supporting their hypothesis, they should include it also in the manuscript and not only make its results available on request.

We described the multivariate linear regression analysis in the Materials and Methods section and shortly summarized the results in the Results sections (lines 278-282). The details of the analysis were added to the supplementary information in Table S4.

  1. I suggest to phrase the statement in lines 23 to 25 more carefully by using terms such as “to be involved” or “to contribute to” instead of “to be a cause”.

We added “to  contribute” as suggested.